# Optimized Omnidirectional High-Reflectance Using Octonacci Photonic Crystal for Thermographic Sensing Applications

**Naim Ben Ali** [1,2,*] [ID]**, Youssef Trabelsi** [2,3]**, Haitham Alsaif** [4] [ID]**, Yasssine Bouazzi** [1,2] **and Mounir Kanzari** [2,5]

1   Department of Industrial Engineering, College of Engineering, University of Ha'il,
    Ha'il City 2440, Saudi Arabia; Y.Bouazzi@UOH.EDU.SA
2   Photovoltaic and Semiconductor Materials Laboratory, National Engineering School of Tunis, University of
    Tunis El Manar, Tunis 1002, Tunisia; ytrabelsi@kku.edu.sa (Y.T.); mounir.kanzari@ipeit.rnu.tn (M.K.)
3   Physics Department, College of Arts and Sciences in Muhail Asir, King Khalid University,
    Abha 61421, Saudi Arabia
4   Department of Electrical Engineering, College of Engineering, University of Ha'il,
    Ha'il City 2440, Saudi Arabia; h.alsaif@uoh.edu.sa
5   Preparatory Engineering Institute of Tunis, University of Tunis, Montfleury, Tunis 1008, Tunisia
*   Correspondence: na.benali@uoh.edu.sa; Tel.: +966-595-569-515

**Abstract:** The transmittance of waves through one-dimensional periodic and Octonacci photonic structures was studied using the theoretical transfer matrix method for both wave-polarization-modes. The first structures were made up of the $SiO_2$ and $TiO_2$ materials. The objective here was to obtain a broad omnidirectional high reflector covering the infrared spectrum of a thermographic camera [1–14 μm] and, especially, to prevent the transmission of emitted human body peak radiation $\lambda_{max} = 9.341$ μm. By comparing the periodic and Octonacci structures, we found that the last structure presented a main and wide photonic band gap near this human radiation. For that, we kept only the Octonacci structure for the rest of the study. The first structure did not give the aspired objective; thus, we replaced the $TiO_2$ layers with yttrium barium copper oxide material, and a significant enhancement of the omnidirectional photonic band gap was found for both TE and TM polarization modes. It was shown that the width of this band was sensitive to the Octonacci iteration number and the optical thickness (by changing the reference wavelength), but it was not affected by the ambient temperature. The number of layers and the thickness of the structure was optimized while improving the omnidirectional high reflector properties.

**Keywords:** omnidirectional; high-reflector; Octonacci; photonic; thermographic; sensing; optimization

## 1. Introduction

Camouflage and the element of surprise are predators' hunting techniques, and for a long time, humans have used them to hide from animals and then surprise and hunt them. Additionally, in military fields, humans use disguise to surprise the enemy, or to take cover from them behind some natural obstacles or behind forts and castles. In recent decades, we have seen some science fiction movies involving camouflage techniques, where humans can walk among people without being noticed. In addition, with the advent of nanotechnology, semiconductors, and metamaterials, we began to notice scientists' efforts to find materials that could be used to prepare clothes which conceal a person from their surroundings. In the field of security, or even surveillance, and military espionage activities, cameras have played an effective role since their advent, along with thermal cameras and night vision devices. Thermographic sensing devices operate within the infrared electromagnetic spectrum.

The infrared spectral band [0.7–30 μm] contains several spectral sub-bands [1]: the near-infrared [0.7–1 μm], the short-wavelength-infrared [1–3 μm], the middle-wavelength-infrared [3–5 μm], the long-wavelength-infrared [8–12 μm], and the very-long-wavelength-infrared [12–30 μm]. Fiber optics, near-infrared spectroscopy, and night vision devices

operate at the near-infrared band. However, for long distances, fiber optics operate at the short-wavelength-infrared band. Infrared guided missiles operate at the middle-wavelength-infrared band. Far-infrared laser and thermal imaging cameras operate at the long-wavelength-infrared band [2]. Nevertheless, the field of work of these cameras expands to include the spectral range 0.9–14 μm [3,4]. These cameras, sometimes-called infrared cameras, detect infrared radiation and produce images of that radiation.

In this study, we searched to design an omnidirectional high reflector using an Octonacci photonic structure which prevents thermal radiation, especially that emitted by humans. These omnidirectional high reflectors could find applications in hiding human thermal infrared radiation and creating a barrier which protects that person from being detected by thermal sensing devices. In addition, these reflectors could hide all thermographic radiation emitted by organisms or machines and find applications in military purposes.

## 2. Theoretical Model

Thermal objects emit infrared radiation which has a maximum emission wavelength ($\lambda_{max}$). This wavelength is inversely proportional to the absolute temperature of the thermal object. Wien's law enables the calculation of $\lambda_{max}$ which represents the peak in the emission spectra from an ideal blackbody [5–7]:

$$\lambda_{max} = \frac{b}{T}$$

where $b$ = 2898 μm.K, which is the Wien's displacement constant [8,9].

Human body temperature is about 310.15 K; therefore, the emitted human body peak radiation (EHBPR) is $\lambda_{max} = 9.341$ μm [8,9].

The studied photonic structures are arranged according to the quasi-periodic Octonacci sequence: $S_n = S_{n-1} * S_{n-2} * S_{n-1}$ [10,11]. Figure 1 shows a schematic of the fourth Octonacci structure which is surrounded by air. The letters H and L represent the layers with high and low refractive index, respectively. The optical responses (the transmittance) are simulated by the transfer matrix method (TMM) [12–15]. In the first part of this study, $SiO_2$ and $TiO_2$ materials were used to build the layers of the Octonacci structure. After that, the yttrium barium copper oxide (YBCO) material replaced the $TiO_2$ layers. To describe the electromagnetic response of the YBCO superconductor at zero external magnetic fields, the Gorter–Casimir two-fluid model was used [12,13].

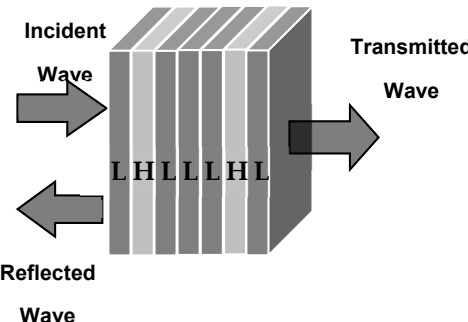

**Figure 1.** Schematic representation showing the geometric structure of the fourth Octonacci iteration.

The refractive index of the YBCO superconductor [12,13] depends to the ambient temperature T and the wave-frequency $\omega$ [16–18]: $n_s = \sqrt{\varepsilon_s} = \sqrt{1 - \frac{1}{\omega^2 \mu_0 \varepsilon_0 \lambda_L^2(T)}}$.

Here, $\lambda_L(T) = \frac{\lambda_{L0}}{\sqrt{1-G(T)}}$ represents the temperature-dependent penetration depth, and $\mu_0$ and $\varepsilon_0$ are the permeability and the permittivity of free space, respectively.

In the last formula, $\lambda_{L0} = 140$ nm is the London penetration depth at T = 0 K, and $G(T) = \left(\frac{T}{T_c}\right)^4$ is the Gorter–Casimir expression [12,13]. In addition, T represents the ambient temperature and $T_c = 92$ K is the superconducting critical temperature [16–18].

## 3. Results and Discussions

### 3.1. Material Effect

First, we began this study by using TiO$_2$ and SiO$_2$ as high (H) and low (L) refractive index materials to construct the photonic structures. By following the Sellmeier equation, the variation of the refractive indices of these materials in the studied spectral range 0.9–14 μm was considered constant. Therefore, the considered refractive indexes of TiO$_2$ and SiO$_2$ were $n_H = 2.3$ and $n_L = 1.45$, respectively [19]. The optical thicknesses $n_{L,H} * d_{L,H}$ of layers L and H were chosen to satisfy the Bragg's condition [19,20]: $n_L * d_L = n_H * d_H = \frac{\lambda_0}{4}$, where $n_{L,H}$, $d_{L,H}$ and $\lambda_0$ are the refractive indices, the geometric thickness of layers and the reference wavelength, respectively. Concerning the incident light, we could have a superposition of different linear, elliptical, and circular polarizations. Natural light is thus made up of many different polarization states. However, although sometimes the sources may be unpolarized, reflection and passing through certain media can alter the polarization of light. Natural light is often represented by a beam of light polarized in two orthogonal directions, and in this study, we consider both the TE and TM polarization of light.

Figures 2 and 3 show the transmittance spectra for both TE and TM wave-polarization and for periodic and Octonacci structures, respectively. In these figures, the dark blue represents the photonic band gap (PBG) for which the transmission coefficient is less than 2%. For both periodic and Octonacci structures, the number of layers NL and the reference wavelength $\lambda_0$ were fixed at 17 and 6.5 μm, respectively. Therefore, the structure thicknesses of periodic and Octonacci structures were 15.32 and 16.98 μm, respectively. For the periodic structure and for both polarization modes, it is clear that the EHBPR ($\lambda_{max} = 9.341$ μm) was allowed to pass through the photonic structure for the majority of the incidence angle $\theta_0$ values (see Figure 2). For the periodic structure, the large and main photonic band gaps (PBGs) were located around the reference wavelength $\lambda_0$. However, for the Octonacci structure (see Figure 3), and due to the quasiperiodic distribution of layers, the presence of several narrow photonic band gaps (PBGs) for both wave-polarization modes was clear. In addition, Figure 3 shows that the main and large PBG was located near the EHBPR ($\lambda_{max} = 9.341$ μm). In fact, Figure 3 shows that for TE mode and from the wavelength of 8.638 μm to 9.675 μm, the wave is transmitted; therefore, the EHBPR ($\lambda_{max} = 9.341$ μm) could pass through the photonic structure for the incidence angle $\theta_0 \leq 19.19$ degree. However, for TM mode the EHBPR ($\lambda_{max} = 9.341$ μm) still transmitted only when the incidence angle $\theta_0 \leq 13.98$ dgree; after that, it was prevented from passing through the structure. Therefore, from the comparison of Figures 2 and 3, we noticed the importance of using the Octonacci structure to prevent the transmission of the EHBPR through the photonic structure.

To expand the PBGs to cover the EHBPR ($\lambda_{max} = 9.341$ μm), the layers of TiO$_2$ materials were replaced with the superconductor yttrium barium copper oxide (YBCO). Figure 4 shows the transmittance spectra as a function of wavelength (μm) and the incident wave angle (radian) for TE and TM wave-polarization. The iteration number, the number of layers, and the reference wavelength were still the same (S = 5, NL = 17, $\lambda_0$ = 6.5 μm). The ambient temperature of the superconductor was 25 °C. From Figure 4, it is clear that the Octonacci structure prevented the transmission of waves for the spectrum [3.48–14 μm], for all wave-incidence angle [0–90 degrees], and for both wave-polarization modes TE and TM. Therefore, this structure functions as a high omnidirectional reflector and the EHBPR ($\lambda_{max} = 9.341$ μm) cannot pass through the photonic structure (it is totally reflected). This physical phenomenon of the widening of the PBG is due to the change of the refractive index contrast when the TiO$_2$ layers have been changed by the YBCO superconductor layers. In reality, the EHBPR is located around the wavelength $\lambda_{max} = 9.341$ μm, and from Figure 4 and when we optimized the structure, the PBG that covered the emitted human body peak was extended and surrounded not only this peak but a large range besides this wavelength.

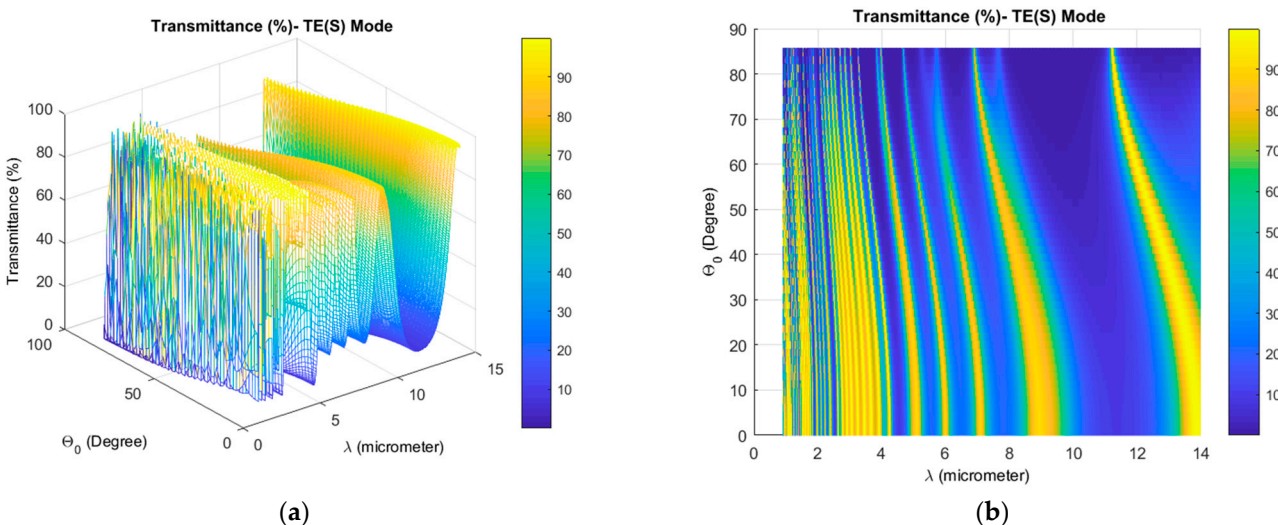

**Figure 2.** Transmittance spectra as a function of wavelength $\lambda (\mu m)$ and wave-incidence angle $\theta_0$ (*degree*) for a 1D periodic multilayered stack (TiO$_2$ and SiO$_2$ materials) with NL = 17, $\lambda_0 = 6.5$ µm and d = 15.32 µm: (**a**) Three-dimensional transmittance for TE mode, (**b**) top view of the 3D transmittance for TE mode, (**c**) Three-dimensional transmittance for TM mode and (**d**) top view of the 3D transmittance for TM mode.

**Figure 3.** *Cont.*

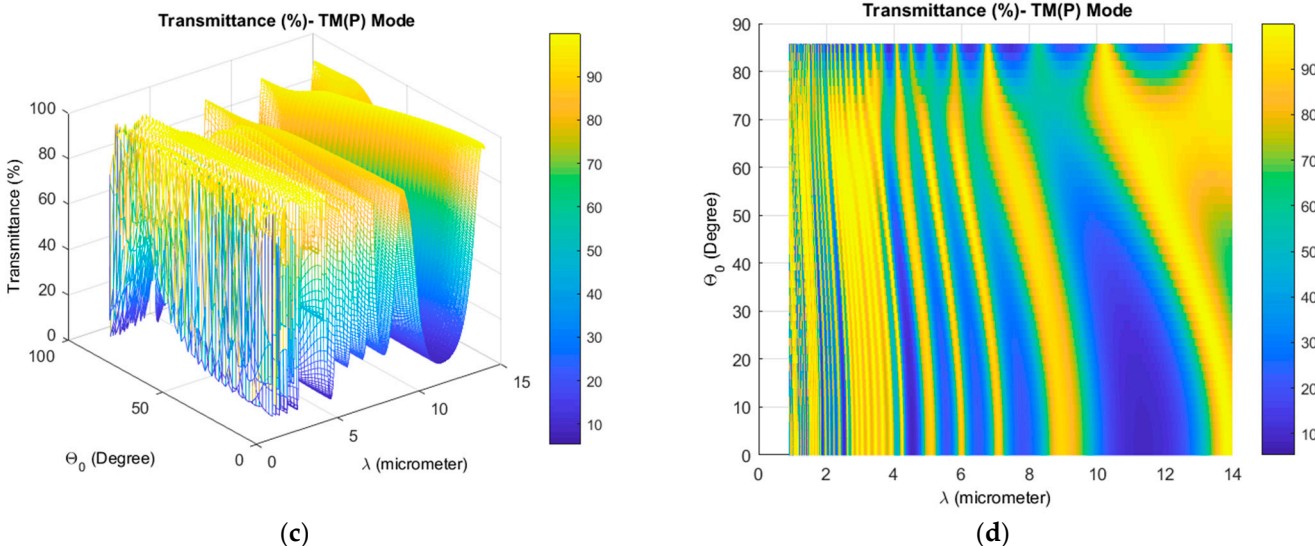

(**c**)　　　　　　　　　　　　　　(**d**)

**Figure 3.** Transmittance spectra as a function of wavelength $\lambda$ (μm) and wave-incidence angle $\theta_0$ (*degree*) for a 1D Octonacci multilayered stack (TiO$_2$ and SiO$_2$ materials) with S = 5, NL = 17, $\lambda_0 = 6.5$ μm and d = 16.98 μm: (**a**) Three-dimensional transmittance for TE mode, (**b**) top view of the 3D transmittance for TE mode, (**c**) Three-dimensional transmittance for TM mode and (**d**) top view of the 3D transmittance for TM mode.

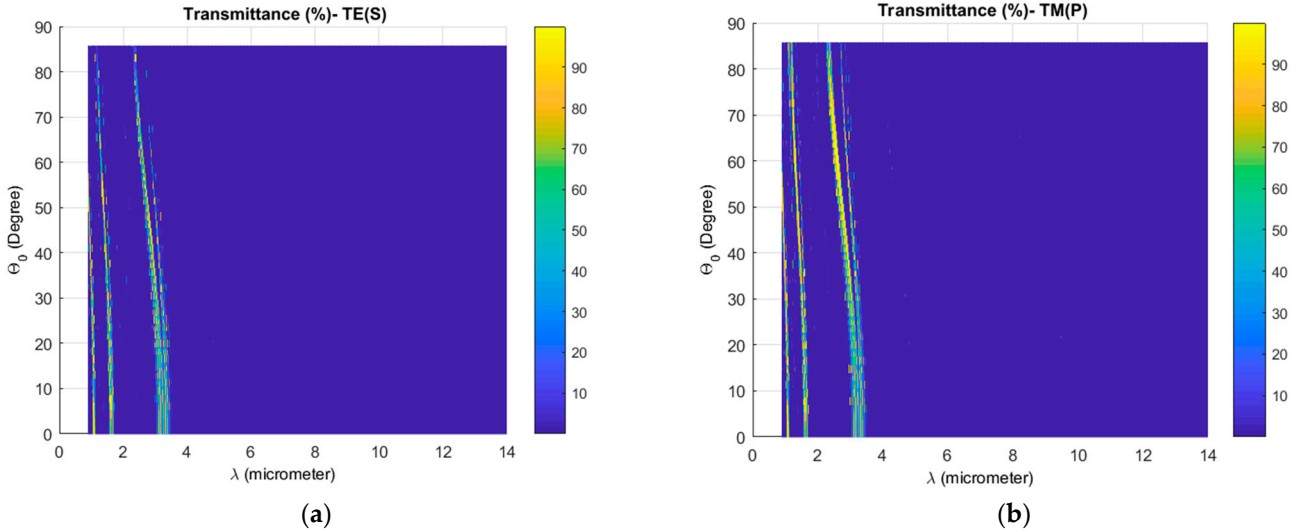

(**a**)　　　　　　　　　　　　　　(**b**)

**Figure 4.** Top view of the 3D transmittance spectra as a function of wavelength $\lambda$ (μm) and wave-incidence angle $\theta_0$ (*degree*) for a 1D Octonacci multilayered stack (with YBCO and SiO$_2$ materials) and for TE and TM wave-polarization modes: S = 5, NL = 17, $\lambda_0 = 6.5$ μm and d = 13.49 μm: (**a**) for TE mode and (**b**) for TM mode.

### 3.2. Iteration Effect

From Figure 4, it can be seen that the Octonacci structure displayed the desired high reflectors for the thermography wavelength spectrum. Next, we searched to optimize the structure thickness by reducing the iteration number of the Octonacci sequence from five to four, so that the number of layers NL was reduced from 17 to 7, and the thickness was $d = 5.62$ μm. Figure 5 illustrates the transmittance spectra as a function of wavelength (μm) and incident wave angle (radian) for TE and TM wave-polarization. The wave was still prevented from transmitting for the wavelength spectrum [3.48–14 μm] and for both polarization modes, and the EHBPR ($\lambda_{max} = 9.341$ μm) was inhibited for the majority of the $\theta_0$ interval. Therefore, when the number of layers and the thickness of the structure were optimized to be $d = 5.62$ μm, we found results close to those given by the previous structure when $d = 13.49$ μm (Figure 4).

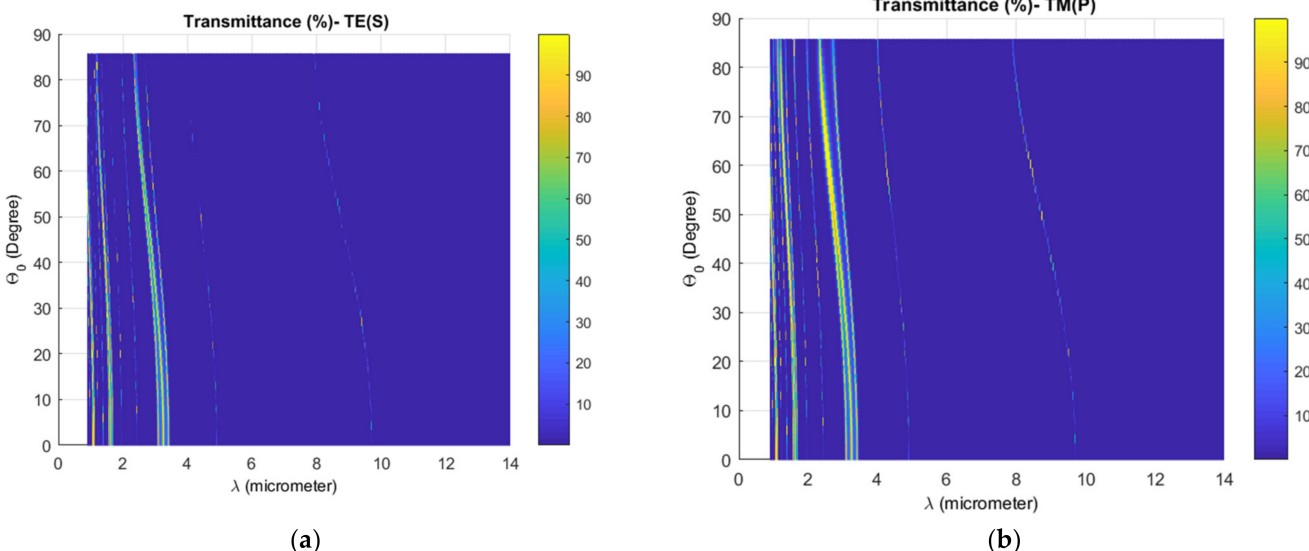

**Figure 5.** Top view of the 3D transmittance spectra as a function of wavelength $\lambda(\mu m)$ and wave-incidence angle $\theta_0$ (*degree*) for the 1D Octonacci multilayered stack (with YBCO and SiO$_2$ materials) and for TE and TM wave-polarization modes: S = 4, NL = 7, $\lambda_0$ = 6.5 μm and d = 5.62 μm: (**a**) for TE mode and (**b**) for TM mode.

### 3.3. Reference Wavelength Effect

The reference wavelength $\lambda_0$ determined the layers' optical and geometric thicknesses ($n_L * d_L = n_H * d_H = \frac{\lambda_0}{4}$) [19,20]; therefore, by changing this wavelength, we again optimized the structure thickness d. The previous parameters of the structure presented in Figure 4 were kept, and we changed only the $\lambda_0$ value. In Figure 6a,b, $\lambda_0$ became equal to 3.5 μm, thus the structure thickness was reduced to d = 3.02 μm. Here, for both polarization modes, it was clear that the transmittance peaks moved toward the lower wavelengths and no transmittance peak intercepted the EHBPR ($\lambda_{max}$ = 9.341 μm) for the whole $\theta_0$ interval. In Figure 6c–h, $\lambda_0$ was changed to become 1.5 μm, 1 μm and 0.5 μm, respectively, and the structure thickness was optimized again to be d = 1.29 μm, d = 0.86 μm and d = 0.43 μm, respectively. The best omnidirectional high reflector for both polarization modes was found when $\lambda_0$ = 1.5 μm (Figure 6c,d). For $\lambda_0$ = 1 μm, the TE mode prevented the EHBPR from transmitting inside the Octonacci structure, but the TM allowed a weak percentage of it to pass through the structure when $\theta_0 \geq 57.3$ *degree* (see Figure 6e,f). Finally, when $\lambda_0$ became equal to 0.5 μm, the transmittance peaks appeared in both polarization modes, which is due to the huge difference between the optical thickness of layers and the incident wavelengths.

The widening of the PBG and the presence of some propagation modes within it is due to the layers' thickness change when the reference wavelength $\lambda_0$ is modified. For the rest of the paper, we will keep $\lambda_0$ = 1.5 μm and d = 1.29 μm.

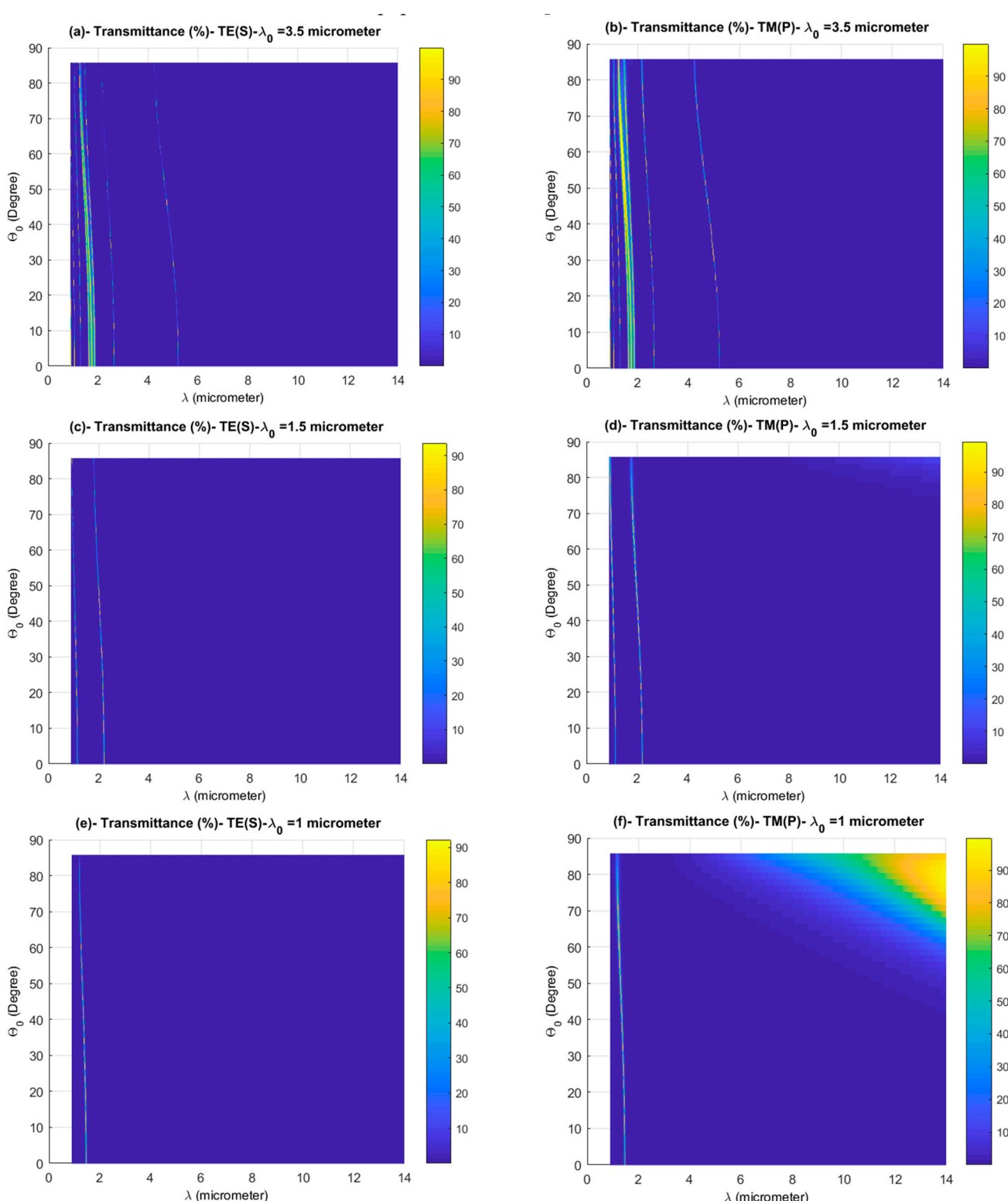

**Figure 6.** *Cont.*

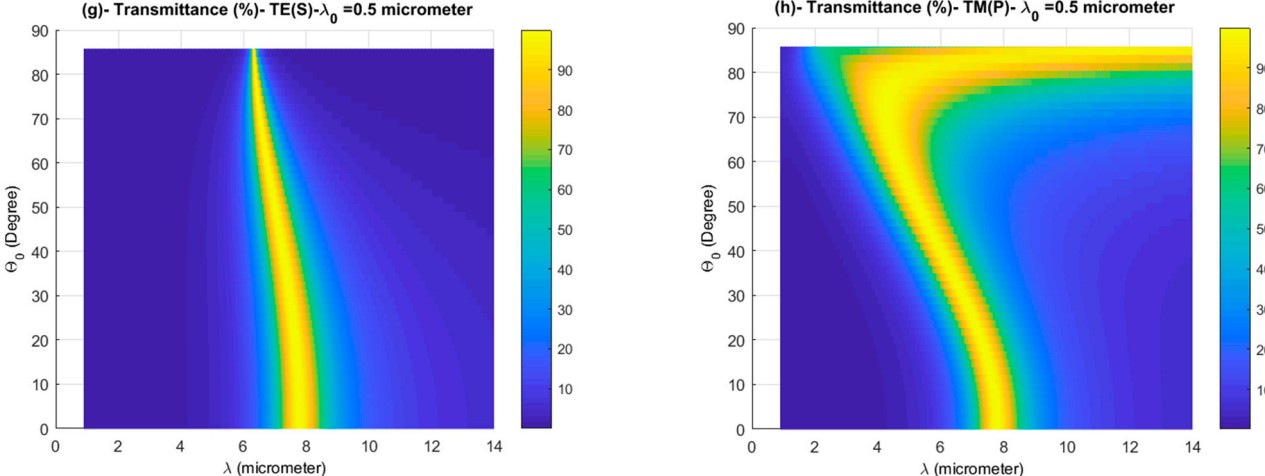

**Figure 6.** Top view of the 3D transmittance spectra as a function of wavelength $\lambda\,(\mu m)$ and wave-incidence angle $\theta_0$ (*degree*) for a 1D Octonacci multilayered stack (with YBCO and SiO$_2$ materials) and for TE and TM wave-polarization modes (S = 4 and NL = 7): (**a**,**b**) $\lambda_0 = 3.5$ μm, (**c**,**d**) $\lambda_0 = 1.5$ μm, (**e**,**f**) $\lambda_0 = 1$ μm, and (**g**,**h**) $\lambda_0 = 0.5$ μm.

### 3.4. Temperature Effect

In this part of the paper, we will present the ambient temperature effect on the wave transmission through the Octonacci structure. The optimized parameters of the photonic structure were: S = 4 for the iteration number, NL = 7 for the number of layers, $\lambda_0 = 1.5$ μm for the reference wavelength, and d = 1.29 μm for the structure thickness. In Figure 7, the ambient temperature is changed from $-20\,°C$ to $60\,°C$ to simulate all the weather conditions encountered by the photonic structure from the lowest temperatures in the world's coldest regions to the highest temperatures in the desert. Figure 7 suggests that the temperature has no effect on the wave transmission inside the Octonacci structure, but from Figure 6a–d, we can observe the presence of a small percentage of transmittance when the temperature is less than $0\,°C$, the $\theta_0 > 57.3$ *degree* and for the highest wavelengths. However, this percentage is still very weak and does not affect blocking the EHBPR. In addition, we can observe the presence of two weak resonance peaks in the low wavelengths; the first one is located near 1 μm, and the second one is located near 2 μm. When the temperature exceeds $20\,°C$, these peaks shift a little towards the higher wavelengths. Therefore, the optimized Octonacci structure represents an omnidirectional high reflector for the infrared spectrum of thermographic sensing, and especially inhibits the EHBPR from transmitting inside it.

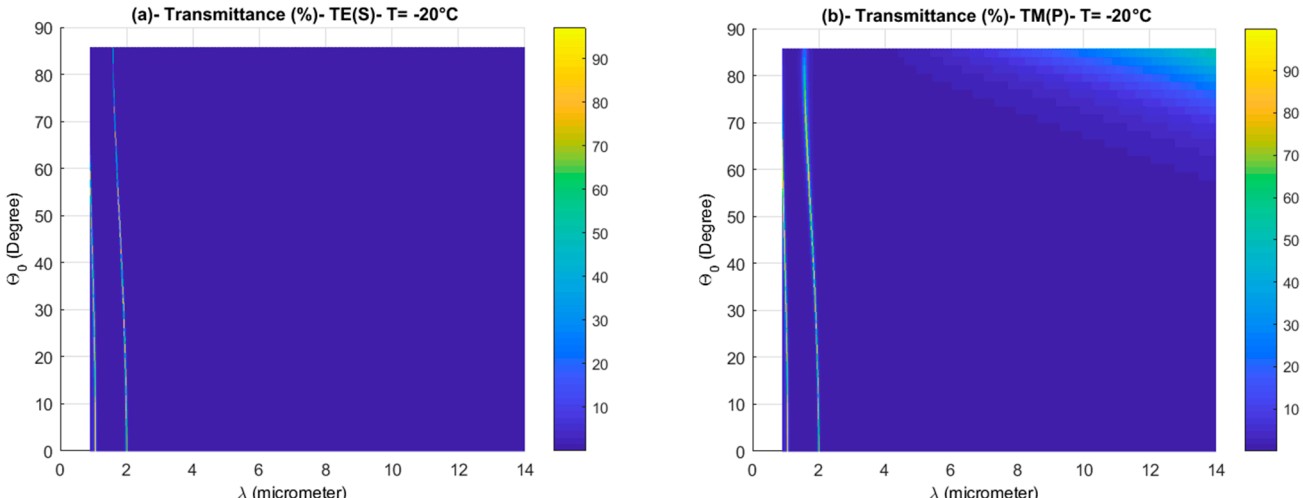

**Figure 7.** *Cont.*

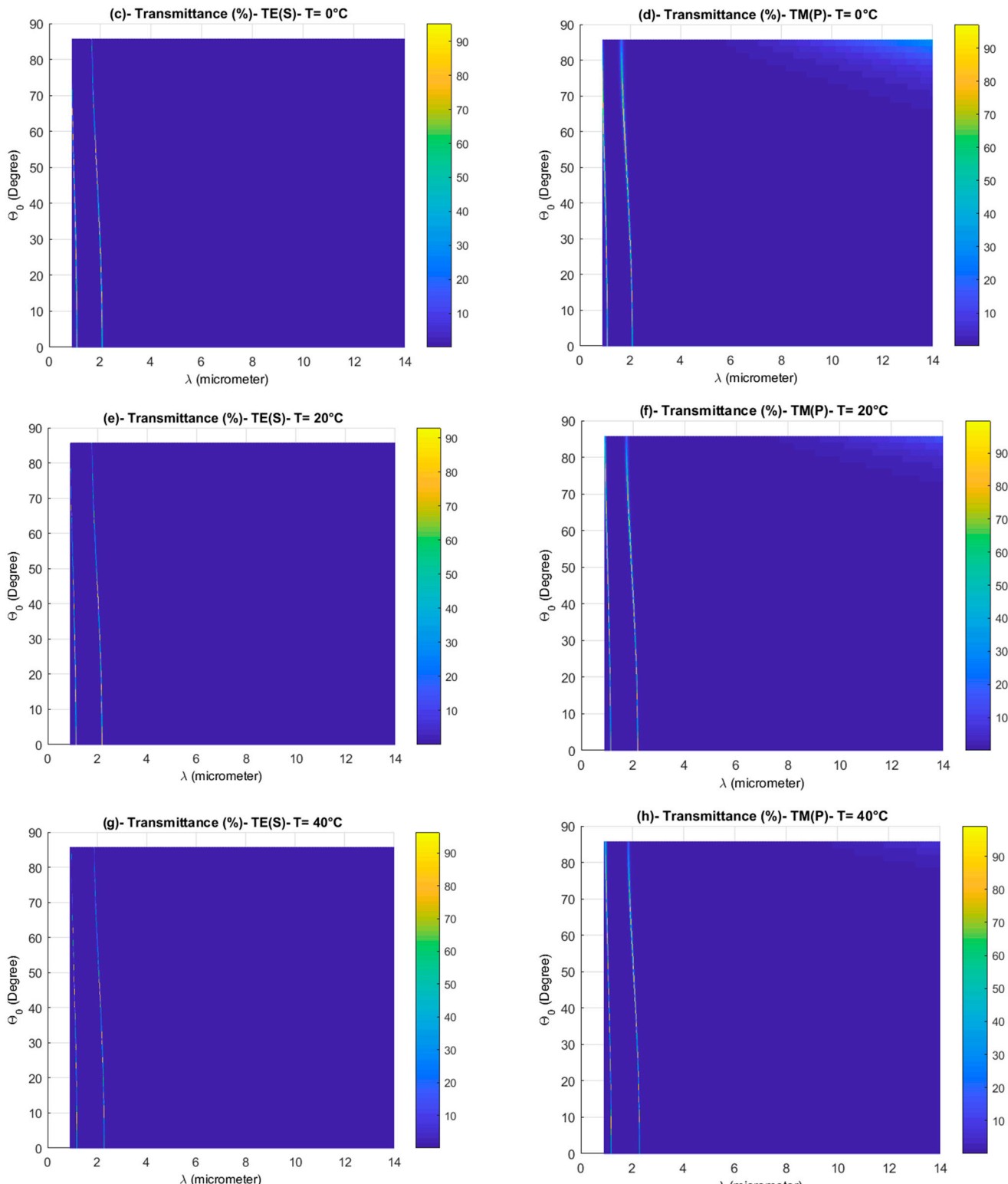

**Figure 7.** *Cont.*

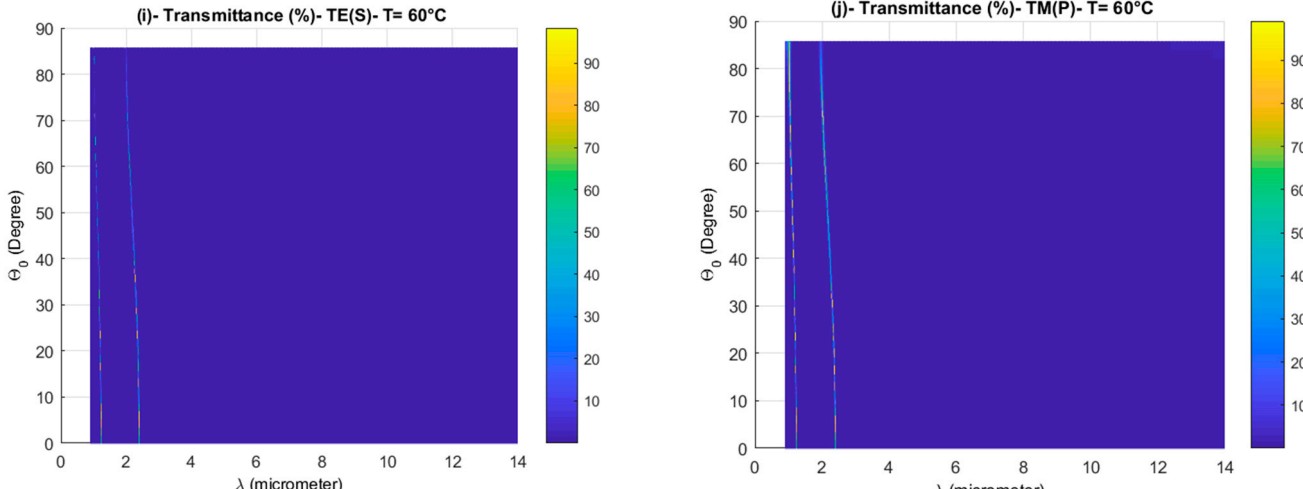

**Figure 7.** Top view of the 3D transmittance spectra as a function of wavelength $\lambda\,(\mu m)$ and wave-incidence angle $\theta_0$ (*degree*) for the 1D Octonacci multilayered stack (with YBCO and SiO$_2$ materials) and for TE and TM wave-polarization modes (S = 4, NL = 7, $\lambda_0$ = 1.5 $\mu$m and d = 1.29 $\mu$m): (**a,b**) T = $-20$ °C, (**c,d**) T = 0 °C, (**e,f**) T = 20 °C, (**g,h**) T = 40 °C, and (**i,j**) T = 60 °C.

## 4. Conclusions

In this paper, the behavior of the PBG of an Octonacci structure is studied. Firstly, we studied periodic and Octonacci structures which were constructed of the materials SiO$_2$ and TiO$_2$. For the periodic structure, the transmittance spectra, showing the presence of a main PBG located near the reference wavelength and the EHBPR ($\lambda$\_max = 9.341 $\mu$m), could transmit through the structure for the majority of $\theta_0$ values. However, for the Octonacci structure, the presence of several narrow PBGs was noticed, as well as a main and large PBG present near the EHBPR, which prevented this radiation for some values of $\theta_0$. We retained only the Octonacci structure for the rest of the study. Secondly, and in order to enlarge the main PBG, the TiO$_2$ material was replaced with the superconductor YBCO. Here, the transmittance spectra for both polarization modes showed the presence of a large, main PBG that covered the infrared spectrum [3.48–14 $\mu$m]. In addition, the EHBPR ($\lambda$\_max = 9.341 $\mu$m) could not transmit through the structure. After that, the thickness of the structure was optimized by changing the iteration number of the Octonacci sequence from five to four, and the thickness of the structure changed from 13.49 $\mu$m to 5.62 $\mu$m. The optical properties were still the same with these optimization parameters. Again, the thickness of the structure was optimized by changing the reference wavelength, and the best omnidirectional high reflector for both polarization modes was found when $\lambda_0$ = 1.5 $\mu$m. Here, the thickness of the structure became 1.29 $\mu$m. Finally, the effects of ambient temperature on the optical properties of the Octonacci reflector were studied. The simulated temperature varied from $-20$ °C to 60 °C, and we did not notice any obvious changes in the main PBG. In fact, the EHBPR was still reflected by the structure (inhibited from transmitting through the Octonacci structure).

**Author Contributions:** Conceptualization, N.B.A. and Y.T.; methodology, H.A.; software, N.B.A. and Y.B.; validation, N.B.A. and M.K.; formal analysis, N.B.A.; investigation, Y.B.; resources, H.A.; data curation, N.B.A.; writing—original draft preparation, N.B.A. and Y.T.; writing—review and editing, N.B.A. and M.K.; visualization, M.K.; supervision, N.B.A.; project administration, N.B.A.; funding acquisition, all authors. All authors have read and agreed to the published version of the manuscript.

**Funding:** This research has been funded by the Scientific Research Deanship at University of Ha'il— Saudi Arabia through project number RG-20 013.

**Institutional Review Board Statement:** Not applicable.

**Informed Consent Statement:** Not applicable.

**Acknowledgments:** This research has been funded by Scientific Research Deanship at University of Ha'il—Saudi Arabia through project number RG-20 013.

**Conflicts of Interest:** The authors declare no conflict of interest. The funders had no role in the design of the study; in the collection, analyses, or interpretation of data; in the writing of the manuscript, or in the decision to publish the results.

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
