# Peer review of "Optimized Omnidirectional High-Reflectance Using Octonacci Photonic Crystal for Thermographic Sensing Applications"

_photonics, doi:10.3390/photonics8050169_

Round 1

Reviewer 1 Report

In this paper, the authors use transfer matrix method to calculate the transmission spectra of 1D periodic and Octonacci photonic structures. There are at least three flaws that should be carefully addressed. First, the refractive indices of SiO2 and TiO2 are not described; besides, the surround medium which the plane waves enter is not clearly defined. Except YBCO, is there any wavelength dependence for the indices of SiO2 and TiO2? Second, only color maps are shown for transmission spectra. In order to determine the forbidden wavelength range (or photonic bandgap), the authors should define the transmission coefficients less than certain value (5% or 2%) as the decision level. Third, the references section is a mess. It neither consistent nor conform to the format. Moreover, several typos and mistakes are listed and should be revised according to the following points.

  1. The London penetration depth (l0) in line 83 and 85 of page 2 is ambiguous and can be confused with the reference wavelength in line 94 of page 3.
  2. The reference wavelength (l0) in line 105 of page 3 is missing.
  3. The “transmitted electric field” is not actually calculated and is redundant in the text (g. line 75 of page 2) as well as in the captions of all figures.
  4. Line 192 of page 7: change “fro” to “for”
  5. I am afraid that a few paragraphs in section 3.2 and 3.3 do not convey the idea and require editing of English language and style.

Author Response

Response to Reviewers’ comments

 May 04, 2021

 Subject: Revised Paper

Manuscript ID: Photonics-1201522

Title: Optimized Omnidirectional High-Reflectance using Octonacci Photonic Crystal for Thermographic sensing applications.

Dear Editor,

With regard to your E-Mail of May 03, 2021, I’m pleased to inform you that the paper has been revised as demanded, the details, the language and the format are seriously improved. I am thankful for the reviewers’ constructive remarks that helped to considerably improve and clarify my paper. As below, I would like to illustrate how I considered the critiques in this updated version. I have addressed the comments as outlined below, in italic font and my responses in blue color and normal font. Additionally, all changes made in the scientific paper have been highlighted in yellow.

Responses to the first reviewer's report:

In this paper, the authors use transfer matrix method to calculate the transmission spectra of 1D periodic and Octonacci photonic structures. There are at least three flaws that should be carefully addressed. First, the refractive indices of SiO2 and TiO2 are not described; besides, the surround medium which the plane waves enter is not clearly defined. Except YBCO, is there any wavelength dependence for the indices of SiO2 and TiO2? Second, only color maps are shown for transmission spectra. In order to determine the forbidden wavelength range (or photonic bandgap), the authors should define the transmission coefficients less than certain value (5% or 2%) as the decision level. Third, the references section is a mess. It neither consistent nor conform to the format. Moreover, several typos and mistakes are listed and should be revised according to the following points.

In the part 2, the refractive indices of SiO2 and TiO2 are described and this sentence is added in the manuscript: By following the Sellmeier equation the variation of the refractive indices of these materials in the studied spectral range 0.9-14µm is considered constant. Therefore the considered refractive index of TiO2 and SiO2 are  and  respectively [19].”.

The surround medium which the plane waves enter is now defined and this sentence is added in page 2: “Figure 1 shows a shematic of the 4th Octonacci structure which is surrounded by air..

In addition, to clarify the photonic band gaps for all graphs these sentence was added: “In these figures the dark blue represents the photonic band gap (PBG) for which the transmission coefficient is less than 2%”.

The references were formatted according to the journal format and the ref. [4] was changed by the new one:

[4] William Frodella, Giovanni Gigli, Stefano Morelli, Luca Lombardi and Nicola Casagli, Landslide Mapping and Characterization through Infrared Thermography (IRT): Suggestions for a Methodological Approach from Some Case Studies, Remote Sens. 2017, 9, pp. 1281; https://doi.org/10.3390/rs9121281

  1. The London penetration depth (l0) in line 83 and 85 of page 2 is ambiguous and can be confused with the reference wavelength in line 94 of page 3.

The symbols of the London penetration depth and the reference wavelength are rectified.

 The reference wavelength (l0) in line 105 of page 3 is missing.

The reference wavelength is inserted.

  1. The “transmitted electric field” is not actually calculated and is redundant in the text (g. line 75 of page 2) as well as in the captions of all figures.

In the transfer matrix method the incident, transmitted and refractive electric field are related via the product matrix. In fact the transmittance and the reflectance are related to the electric fields and for that in all captions the 2D graph represents the top view of the transmittance. The captions of all figures are changed.

  1. Line 192 of page 7: change “fro” to “for”

This typo is corrected.

  1. I am afraid that a few paragraphs in section 3.2 and 3.3 do not convey the idea and require editing of English language and style.

The content of the discussion was enriched, the language and the format of the manuscript are seriously improved.

Finally, if you have any remarks, please do not hesitate to contact me and thank you very much for your cooperation.

Reviewer 2 Report

The manuscript report on the modelling of a quasi-random planar photonic crystal without angular dispersion of the photonic band gap  for the shielding of human IR emission. The seems well performed but more details should be provided on the TMM formalism employed.  Overall the manuscript should be improved.

A schematic of the structure could be easier to interpret instead of table 1.

Could the angle be reported in degrees? Radian are not largely employed in the field and data interpretation is not immediate. Also, please report the wavelength in microns or nanometers.

The 3D plot are un-readable. The authors could select a couple of spectra (e.g. normal incidence and 60 degree) and show the stacked instead of those plots.

Regarding the transmittance of the YBCO sample in fig. 3. Cotour plot are usefull to highlight spectral variations. In this case theyare not very meaningful. Would it be possible to add the spectrum (T Vs wavelength )for the sample? Similar comment stands for fig 4.

The author focuses on 9.341μ? but I guess the emission has a quite broad width? Have the author consider it?  

Also, considering the application envisaged, wouldn’t it be more meaningful to show also unpolarized light transmittance?

Regarding fig.4 e relative comments, is d the thickness of the stack. d normally intends the single layer.

Also, Also in this case I am not sure what the contour plot should show? Can spectra be reported as well?

In Figure 6 I would say that there are not major differences between the different temperatures but it is possible to clearly distinguish at least some intensity difference. Comparing  the spectra at normal incidence could help visualizing analogies and differences. May be all these contours could be moved to SI.

There are typos and different fonts employed along the manuscript.

Author Response

Response to Reviewers’ comments

May 04, 2021

 Subject: Revised Paper

Manuscript ID: Photonics-1201522

Title: Optimized Omnidirectional High-Reflectance using Octonacci Photonic Crystal for Thermographic sensing applications.

Dear Editor,

With regard to your E-Mail of May 03, 2021, I’m pleased to inform you that the paper has been revised as demanded, the details, the language and the format are seriously improved. I am thankful for the reviewers’ constructive remarks that helped to considerably improve and clarify my paper. As below, I would like to illustrate how I considered the critiques in this updated version. I have addressed the comments as outlined below, in italic font and my responses in blue color and normal font. Additionally, all changes made in the scientific paper have been highlighted in yellow.

Responses to the second reviewer's report:

The manuscript report on the modelling of a quasi-random planar photonic crystal without angular dispersion of the photonic band gap  for the shielding of human IR emission. The seems well performed but more details should be provided on the TMM formalism employed.  Overall the manuscript should be improved.

A schematic of the structure could be easier to interpret instead of table 1.

We added a the new figure 1 instead of table 1 to interpret the structure 1

Could the angle be reported in degrees? Radian are not largely employed in the field and data interpretation is not immediate. Also, please report the wavelength in microns or nanometers.

The angle is now reported in degree and the wavelength in micron.

The 3D plot are un-readable. The authors could select a couple of spectra (e.g. normal incidence and 60 degree) and show the stacked instead of those plots.

The objective of the paper is to achieve an omnidirectional reflector that cavers the emitted human body peak radiation, therefore an omnidirectional means the range of incidence angle [0-π/2]. The 2D graph represents the top view of the transmittance. The resolution of all figures is improved. The presence of the transmission peaks is highlighted with yellow color and the photonic bad gaps with dark blue color.

Regarding the transmittance of the YBCO sample in fig. 3. Cotour plot are usefull to highlight spectral variations. In this case theyare not very meaningful. Would it be possible to add the spectrum (T Vs wavelength) for the sample? Similar comment stands for fig 4.

For figure 3 and 4, the structure presents a wide photonic bang gap (highlighted with the dark blue color) which surround the emitted human body peak radiation, therefore we focused on this zone and we improved the resolution of these two figure to show this zone clearly.

The author focuses on 9.341μ? but I guess the emission has a quite broad width? Have the author consider it?  

From figure 4 and when we optimized the structure the photonic band gap that cover the emitted human body peak is extended and surround not only this peak but a large range beside this wavelength.  In page 6 and just before figure 4, this sentence is added: “In reality the the EHBPR locates around the wavelength  and from figure 4 and when we optimized the structure, the PBG that cover the emitted human body peak is extended and surround not only this peak but a large range beside this wavelength.”.

Also, considering the application envisaged, wouldn’t it be more meaningful to show also unpolarized light transmittance?

To respond to this question a new paragraph is added in part 3.1 (page 3): “Concerning the incident light we can have a superposition of different linear, elliptical and circular polarizations. Natural light is thus made up of many different polarization states. However, although sometimes the sources may be unpolarized, reflection and passing through certain media can alter the polarization of light. Natural light is often represented by a beam of light polarized in two orthogonal directions and in this study we consider both the polarization of light TE and TM.”.

Regarding fig.4 e relative comments, is d the thickness of the stack. d normally intends the single layer.

Yes d represents the thickness of the whole structure.

Also, Also in this case I am not sure what the contour plot should show? Can spectra be reported as well?

The 2D graph represents the top view of the transmittance. The resolution of all figures is improved. The presence of the transmission peaks is highlighted with yellow color and the photonic bad gaps with dark blue color.

In Figure 6 I would say that there are not major differences between the different temperatures but it is possible to clearly distinguish at least some intensity difference. Comparing  the spectra at normal incidence could help visualizing analogies and differences. May be all these contours could be moved to SI.

We have added in page 10 and just before figure 7 this sentence to better compare the spectra: “In addition, we can notice the presence of two weak resonance in the low wavelegnths; the first one locates near 1 micron and the second one locates near 2 micron and when the temperature excedds 20°C, these peaks these peaks shift a little towards the high wavelengths.”.

There are typos and different fonts employed along the manuscript.

All typos are revised and corrected.

Finally, if you have any remarks, please do not hesitate to contact me and thank you very much for your cooperation.

Round 2

Reviewer 1 Report

Attached please find the comments.

Author Response

Response to Reviewers’ comments

May 10, 2021

Subject: Revised Paper

Manuscript ID: Photonics-1201522

Title: Optimized Omnidirectional High-Reflectance using Octonacci Photonic Crystal for Thermographic sensing applications.

Dear Editor and reviewers,

With regard to your E-Mail of May 10, 2021, I’m pleased to inform you that the paper has been revised as demanded, the details, the language and the format are seriously improved. I am thankful for the reviewers’ constructive remarks that helped to considerably improve and clarify my paper. As below, I would like to illustrate how I considered the critiques in this updated version. I have addressed the comments as outlined below, in italic font and my responses in blue color and normal font. Additionally, all changes made in the scientific paper have been highlighted in yellow.

Responses to the first reviewer's report:

The manuscript is much improved after the revision. However, several typos and mistakes still appear and listed as follows,

  1. The wavelength unit is micrometer instead of micron; moreover, the unit in the figure caption should be corrected as well. Please check carefully from Figure 2 to Figure 7.

This typo was corrected in all figures.

  1. The “transmitted electric field” is only valid when you define the input electric field as unit one. The transmittance spectra are more precise since you only calculate their ratios. I think the “transmitted electric field” is redundant in the text (e.g. line 67 of page 2) as well as in the captions of all figures.

This mistake was corrected in the text and in all figures captions.

  1. In Figure 6 and Figure 7, there are two (a), two (b), two (c) and two (d). I do not think it is a good idea.

All figures were rectified as demanded.

  1. The “weather temperature” in line 220 of page 12 is not precise. It is better to use “ambient temperature”.

The word was changed as demanded.

Moreover, I am afraid that a few paragraphs do not convey the idea and require editing of English language and style. For example,

  1. Line 147-149 of page 7, “we only notice the presence of weak transmitted electric field …… when q0 = 27.96 degree”. I cannot find its logical relation in the context.

This sentence is deleted and the paragraph is rewritten to be: “The wave is still forbidden to transmit for the wavelength spectrum [3.48- 14] and for both polarization modes and the EHBPR () is banned for the majority of  intervals.”.

  1. Line 171-174 of page 8, “The physical phenomenon of change of the width of the PBG and of the presence of some propagating modes ……” It is difficult to understand since there are 7 “of” in this sentence.

This sentence is rewritten and the new one is: “The widening of the PBG and the presence of some propagation modes within is due to the layers’ thickness change when the reference wavelength is modified.”.

Finally, if you have any remarks, please do not hesitate to contact me and thank you very much for your cooperation.

Best regards,

Reviewer 2 Report

The Authors replied appropriately to all my concerns and the paper is now suitable for publication in the journal. 

Author Response

Response to Reviewers’ comments

 May 10, 2021

 Subject: Revised Paper

Manuscript ID: Photonics-1201522

Title: Optimized Omnidirectional High-Reflectance using Octonacci Photonic Crystal for Thermographic sensing applications.

Dear Editor and reviewers,

With regard to your E-Mail of May 10, 2021, I’m pleased to inform you that the paper has been revised as demanded, the details, the language, and the format are seriously improved. I am thankful for the reviewers’ constructive remarks that helped to considerably improve and clarify my paper. As below, I would like to illustrate how I considered the critiques in this updated version. I have addressed the comments as outlined below, in italic font and my responses in blue color and normal font. Additionally, all changes made in the scientific paper have been highlighted in yellow.

Responses to the second reviewer's report:

Thank you for reviewing the paper and your comments were helpful.

Finally, if you have any remarks, please do not hesitate to contact me and thank you very much for your cooperation.

Best regards,

This manuscript is a resubmission of an earlier submission. The following is a list of the peer review reports and author responses from that submission.

Round 1

Reviewer 1 Report

The file is attached
